# Risk Factors for Non-Communicable Diseases at Baseline and Their Short-Term Changes in a Workplace Cohort in Singapore

**DOI:** 10.3390/ijerph16224551

**Published:** 2019-11-18

**Authors:** Thirunavukkarasu Sathish, Gerard Dunleavy, Michael Soljak, Nanthini Visvalingam, Nuraini Nazeha, Ushashree Divakar, Ram Bajpai, Thuan-Quoc Thach, Kei L Cheung, Hein de Vries, Chee-Kiong Soh, Georgios Christopoulos, Josip Car

**Affiliations:** 1Centre for Population Health Sciences, Lee Kong Chian School of Medicine, Nanyang Technological University, 11 Mandalay Road, Singapore 308232, Singapore; gerard.dunleavy@ntu.edu.sg (G.D.); m.soljak@imperial.ac.uk (M.S.); nv@rainmaking.io (N.V.); nuraini.nazeha@ntu.edu.sg (N.N.); ushashree.divakar@iqvia.com (U.D.); r.bajpai@keele.ac.uk (R.B.); thuanquoc.thach@ntu.edu.sg (T.-Q.T.); josip.car@ntu.edu.sg (J.C.); 2Population Health Research Institute, McMaster University, 237 Barton Street East, Hamilton, ON L8L 2X2, Canada; 3Department of Health Promotion, CAPHRI Care and Public Health Research Institute, Maastricht University, 6200 MD Maastricht, The Netherlands; hein.devries@maastrichtuniversity.nl; 4Department of Primary Care & Public Health, School of Public Health, Imperial College, London W6 8RP, UK; 5Research Institute for Primary Care & Health Sciences, Keele University, Staffordshire ST5 5BG, UK; 6Department of Clinical Sciences, College of Health and Life Sciences, Brunel University London, London UB8 3PH, UK; KeiLong.Cheung@brunel.ac.uk; 7School of Civil and Environmental Engineering, College of Engineering, Nanyang Technological University, 50 Nanyang Avenue, Singapore 639798, Singapore; CSOHCK@ntu.edu.sg; 8Division of Leadership, Management and Organisation, Nanyang Business School, College of Business, Nanyang Technological University, 50 Nanyang Avenue, Singapore 639798, Singapore; CGeorgios@ntu.edu.sg; 9Decision, Environmental and Organizational Neuroscience Lab, Culture Science Institute, Nanyang Business School, Nanyang Technological University, 50 Nanyang Avenue, Singapore 639798, Singapore; 10Global Digital Health Unit, Department of Primary Care and Public Health, School of Public Health, Imperial College London, London SW7 2AZ, UK

**Keywords:** cohort study, workplace, chronic disease, risk factors, Singapore

## Abstract

We aimed to examine the behavioural and clinical risk factors for non-communicable diseases (NCDs) at baseline and their changes over 12 months in a workplace cohort in Singapore. A total of 464 full-time employees (age ≥ 21 years) were recruited from a variety of occupational settings, including offices, control rooms, and workshops. Of these, 424 (91.4%) were followed-up at three months and 334 (72.0%) were followed up at 12 months. Standardized questionnaires were used to collect data on health behaviours and clinical measurements were performed by trained staff using standard instruments and protocols. Age-adjusted changes in risk factors over time were examined using generalized estimating equations or linear mixed-effects models where appropriate. The mean age of the participants at baseline was 39.0 (SD: 11.4) years and 79.5% were men. Nearly a quarter (24.4%) were current smokers, slightly more than half (53.5%) were alcohol drinkers, two-thirds (66%) were consuming <5 servings of fruit and vegetables per day, and 23.1% were physically inactive. More than two-thirds (67%) were overweight or obese and 34.5% had central obesity. The mean follow-up was 8.6 months. After adjusting for age, over 12 months, there was a significant increase in the proportion consuming <5 servings of fruit and vegetables per day by 33% (*p* = 0.030), who were physically inactive by 64% (*p* < 0.001), and of overweight or obese people by 15% (*p* = 0.018). The burden of several key NCD risk factors at baseline was high and some worsened within a short period of time in this working population. There is a need for more targeted strategies for behaviour change towards a healthy lifestyle as part of the ongoing health and wellness programs at workplaces in Singapore.

## 1. Introduction

Non-communicable diseases (NCDs) are the leading cause of mortality globally. An estimated 40.5 million (71%) of the 56.9 million deaths worldwide in 2016 were due to NCDs [1]. The burden of NCDs is rapidly increasing in both developing and developed countries, which will have significant social, economic, and health consequences unless preventive measures are taken [2]. In Singapore, a developed nation in Southeast Asia, people have been exposed to significant changes in diet and physical activity over the last few decades, a consequence of rapid industrialization and economic growth [3]. This has resulted in a high burden of NCDs and their risk factors in Singapore. For example, it has been estimated that there were over 600,000 adults with diabetes (13.7% prevalence) in 2017 in Singapore [4]. More worryingly, the prevalence of diabetes is projected to increase in the coming years along with the burden of obesity, which is the major driver of the diabetes epidemic [5]. These alarming figures clearly indicate the need for effective measures to control the growing NCD epidemic in the country. 

Health and wellness programs at workplaces can positively impact the health profile of a substantial proportion of a country’s population [6]. Workplace health promotion can also be beneficial to employers due to the economic benefits resulting from reduced absenteeism and increased productivity [7]. Singaporeans spend a significant amount of their day at workplaces [8], thus the workplace is a key setting for health promotion. To control the NCD epidemic in Singapore, the Health Promotion Board has rolled out a variety of initiatives at workplaces such as the “National Steps Challenge—Corporate Challenge” and “Eat. Spin. Win” [9]. There are also smoking cessation programmes, workplace health packages for small and medium enterprises, mental health programmes, and group fitness classes for the working population [10]. To evaluate the effects of such initiatives on employees’ health, it is essential to examine the risk factors and monitor their changes over time [11]. We established a unique workplace cohort in Singapore from across a variety of occupational settings including offices, workshops, and control rooms [12]. In this paper, we aimed to examine the behavioural and clinical NCD risk factors at baseline and their changes over 12 months in this workplace cohort.

## 2. Materials and Methods 

### 2.1. Study Design and Participants

In 2016, a workplace cohort was set up in Singapore to study the health effects of working in underground spaces. Details of the cohort study design are reported elsewhere [12]. Figure 1 shows the participant enrolment and follow-up flowchart. A total of 27 companies from a variety of sectors (transport, banks, universities, learning centres, mail service centres, libraries, cooling plants, and hospitals) in Singapore with underground workspaces were identified by online searches and collaborator referrals. These companies were contacted through personal visits, phone calls, and emails. Fifteen companies (55.6%) were either not willing or not reachable and eight (29.6%) were small with less than 20 employees. The remaining four companies were recruited, which included two transport industries, a cooling plant, and a university. Participants from these companies were invited to participate in the study via meetings, workplace posters, and emails. Those who were willing to participate were screened for eligibility using the following criteria: aged 21 years and above, should be working for at least four hours per day at a particular work location, not pregnant, and had not travelled overseas across a different time zone at least once a month over the past six months. The final component of the eligibility criteria is related to the primary objective of the cohort study, which was to study the effects of working in underground spaces on sleep quality and melatonin levels. Frequent travel across different time zones could potentially influence these outcomes due to circadian disruption [13]. Participants were recruited between 16 August 2016 and 13 January 2017. After three months from baseline, 424 (91.4%) participants were followed-up, and after 12 months from baseline, 334 (72.0%) were followed-up. Of 130 participants who were lost to follow-up at either three or 12 months, 63 (48.5%) were not willing, 30 (23.1%) were not reachable, 23 (17.7%) had left the current job, 11 (8.5%) had moved to a different worksite, and three (2.3%) were pregnant. 

### 2.2. Measurements

The measurements have been reported in detail elsewhere [12]. Briefly, at baseline and follow-up assessments, the same standardized self-report questionnaires were used to collect data on socio-demographic characteristics (age, sex, education, occupation, ethnicity, marital status, and monthly income), work-related characteristics (work years in the present company, work hours per week, work location [underground or aboveground], and shift work), and health behaviours (smoking, alcohol use, diet, and physical activity). Anthropometric measurements (height, weight, and waist and hip circumferences) were performed by trained staff in accordance with standard protocols and tools [11].

### 2.3. Definitions 

#### 2.3.1. Behavioural Risk Factors

Those who were currently smoking any tobacco products (cigarettes, beedies, cigars, or hookah) were defined as current smokers. Alcohol users were those who had at least one standard drink of alcohol in the last 12 months. One standard drink of alcohol equals 30 mL of spirits, 285 mL of beer, or 120 mL of wine [11]. Dietary habits were assessed by a food frequency questionnaire (FFQ), which was adapted from the FFQ that was used in the National Nutrition Survey in Singapore [14]. The FFQ included questions about the usual intake of a range of food items and drinks including rice, porridge, fruits, vegetables, soft drinks, milk and beverages, eggs, seafood, meat and poultry, fried foods, snacks, and nuts over the last 12 months. Standard portion size and four possible frequency of intake responses (daily, weekly, monthly, or never) were given for each food item, which were used to estimate servings per day for those items. We calculated the Alternate Healthy Eating Index-2010 (AHEI-2010) score [15], which is a measure of diet quality, with higher scores being associated with a lower risk of cardiovascular diseases (CVDs), diabetes, and certain cancers [16]. We estimated the AHEI-2010 score using the criteria by Chiuve et al. [15], based on fruits and vegetables, red meat, seafood, sugar-sweetened beverages and fruit juice, nuts and legumes, and brown rice. We did not include alcoholic drinks, sodium intake, and polyunsaturated fats in the calculation of the AHEI score as data for these variables were not available. Each AHEI-2010 component was scored on a scale from 0 (worst) to 10 (best) and the total AHEI-2010 score was calculated by summing the score of individual components. We also estimated the average servings of fruit and vegetables consumed in a day and categorized participants into those meeting the World Health Organization (WHO) recommendation (i.e., consuming ≥5 servings per day) and those that did not (i.e., <5 servings per day) [11]. One serving of fruit or vegetables equals 80 g [11]. Physical activity and sedentary behaviour were measured using the Global Physical Activity Questionnaire (GPAQ) [17]. GPAQ measures activity levels in three domains, namely, work, travel, and leisure. Participants who met the WHO recommendation of at least 150 min of moderate physical activity per week or at least 75 min of vigorous physical activity per week were considered to be active and those that did not meet the recommendation were considered inactive [18]. Sedentary behaviour was defined as the time spent (hours) sitting or reclining in a day. 

#### 2.3.2. Clinical Risk Factors

Trained staff performed clinical measurements such as height, weight, and waist and hip circumferences, according to a standard protocol using standard tools [11]. Height was measured using a stadiometer (Seca 217, Hamburg, Germany) to the nearest 0.1 cm and weight was measured in light clothing using a digital scale (Seca 874, Hamburg, Germany) to the nearest 0.1 kg. Waist and hip circumferences were measured using a stretch-resistance tape (Seca 201, Hamburg, Germany). Waist circumference was measured at the midpoint between the lower margin of the last palpable rib and the top of the iliac crest (hip bone). Hip circumference was measured at the maximum circumference over the buttocks. Body mass index (BMI) was calculated as weight in kilograms divided by the square of height in meters (kg/m^2^). Overweight or obesity was defined as BMI ≥ 23.0 kg/m^2^ as per the WHO Asia-Pacific guidelines for Asian populations [19]. We defined central obesity based on a waist-to-hip ratio of ≥0.90 in men and ≥0.85 in women [20]. 

### 2.4. Statistical Analysis

Baseline characteristics of participants were summarized using mean (SD: standard deviation) or median (IQR: inter-quartile range) for continuous variables and using frequency and percentage for categorical variables. To examine the changes in continuous variables over time, we fitted generalized estimating equations (GEE) with gaussian family and identity link for normally distributed variables, and gamma family and log link for skewed variables with an exchangeable correlation structure. For variables where the estimates were diverging, we fitted linear mixed-effects regression models with maximum likelihood parameter estimation. Timepoint was specified as the fixed effect, and a random effect was specified for individuals to account for correlation between the repeated measurements on the same individual. Age was included as a covariate in all the models. To analyse categorical variables, we used GEE models with binomial family and logit link with an exchangeable correlation structure, and age was included as a covariate in the models. All p values were based on two-tailed tests of significance and those less than 0.05 were considered to be statistically significant. To examine the possibility of selection bias due to loss to follow-up, the baseline characteristics of participants who were followed-up and those who were lost to follow-up were compared using t-test (for normally distributed variables), Wilcoxon rank-sum test (for skewed variables), and χ^2^ test (for categorical variables). Data were analysed with Stata software (version 15.0; Stata Corp LP, College Station, TX, USA).

### 2.5. Ethics Approval 

The study was conducted in accordance with the Declaration of Helsinki and the protocol was approved by the Institutional Review Board of Nanyang Technological University, Singapore (IRB-2015-11-028). Written informed consent was obtained from all the study participants before the commencement of data collection. 

## 3. Results

Table 1 shows the socio-demographic and work-related characteristics of participants at baseline. The mean age was 39.0 (SD: 11.4) years and 79.5% were men. Reflecting the national population, a large proportion was Chinese (63.8%) followed by Malays (21.3%), Indians (10.3%), and other Asian groups (4.5%). The majority (72.0%) were engineers, technicians, or traffic controllers. Nearly half (48.5%) were office staff, 30.2% were control room staff, and 21.3% were workshop staff. Slightly less than one-third (30.6%) were working in underground spaces and 35.8% were shift workers. On average, the study participants were working for 43 h per week. 

### 3.1. Behavioural and Clinical Risk Factors at Baseline

Table 2 shows the behavioural and clinical risk factors for NCDs at baseline. Nearly a quarter (24.4%) were current smokers and were physically inactive (23.1%). Slightly more than half (53.5%) were alcohol drinkers, of which 35.1% had drunk on a regular basis (at least once a month) in the last 12 months. Two-thirds (66%) were consuming fruit and vegetables below the WHO recommended levels i.e., <5 servings per day. More than two-thirds (67%) were either overweight or obese and slightly more than one-third (34.5%) were centrally obese. 

### 3.2. Changes in Risk Factors Over Three and 12 Months

Participants were followed-up for a mean period of 8.6 months (range: 0 to 16.8 months). Participants who were lost to follow-up were significantly younger and more likely to be Malay, employed for a lesser number of years in the current company, and current smokers (Appendix A).

Table 3 shows the age-adjusted changes in behavioural and clinical risk factors for NCDs over three and 12 months. The mean level of total physical activity in a week reduced significantly by 240.7 min (95% CI: −407.4 to −74.1) over three months and by 290.3 min (95% CI: −459.7 to −120.8) over 12 months. Consequently, the proportion who were physically inactive increased significantly by 40% over three months and by 64% over 12 months. The proportion consuming < 5 servings of fruit and vegetables per day increased significantly by 14% over three months and by 33% over 12 months. On average, participants gained 0.23 kgs (95% CI: 0.03 to 0.43) over three months and 0.50 kgs (95% CI: 0.25 to 0.75) over 12 months. BMI increased significantly by 0.08 kg/m^2^ (95% CI: 0.01 to 0.14) over three months and by 0.15 kg/m^2^ (95% CI: 0.06 to 0.23) over 12 months. Consequently, the proportion who were overweight or obese increased significantly by 15% over 12 months.

After adjusting for age, the mean number of risk factors (current smoking, alcohol drinking, physically inactive, intake of <5 servings of fruit and vegetables per day, overweight or obesity, and central obesity) increased from 2.68 (SD: 1.15) to 2.83 (SD: 1.23) over 12 months (*p* = 0.002). There were no significant changes in the proportion of current smokers, alcohol drinkers, or those with central obesity, and in the mean AHEI-2010 score or sedentary behaviour either over three or 12 months.

## 4. Discussion

Most NCDs are due to four key behavioural risk factors (tobacco use, harmful use of alcohol, low physical activity, and unhealthy diet) and four key clinical risk factors (high blood pressure, high blood glucose, overweight or obesity, and high cholesterol) [2]. In this study, we found that the prevalence of some of these risk factors including current smoking, alcohol use, low intake of fruit and vegetables (<5 servings per day), physical inactivity, overweight or obesity, and central obesity at baseline was high in a working population in Singapore. Some of these risk factors, including physical activity levels and the intake of fruit and vegetables (<5 servings per day) reduced, and measures of obesity increased significantly over a period of 12 months. 

The reduction in physical activity levels in our study is in line with findings from previous cohort studies [21] and repeat cross-sectional surveys that have been conducted in the Asian region [22,23]. However, the magnitude of the reduction varied across studies. For example, in the National Health Surveillance Surveys (NHSS) conducted among residents aged 18–69 years in Singapore from 2007 to 2013, the annual increase in the prevalence of physical inactivity was 7% (versus 64% in our study) [22]. This difference could be partly due to the variation in age and sex distribution, the nature of the study population, study methodology, physical activity measurement tools, and the definition of physical inactivity between these studies. However, this could also highlight the high-risk nature of working groups for reduced physical activity due to the transition from manual labour to desk-based jobs, resulting from globalization and rapid economic growth [24]. 

Previous studies have shown that physical activity levels and fruit and vegetable intake decline over time in the absence of any intervention [21,25]. In Singapore, the culture of eating out is very common [14], which puts the population at risk of consuming high-calorie foods without paying attention to the nutritional content. This is reflected by the finding that the quality of diet consumed by our study participants was low with a mean AHEI-2010 score ranging from 23 to 24, at all three time points, out of a maximum possible score of 60. Recent research has shown that frequently ordered food items in restaurants are far more energy-dense than individual requirements for single eating occasions [26]. Food items that are sold in most eateries in Singapore are not usually served with fruit, and the vegetable component in local food choices is also often small in portion size. Only slightly more than one-third (34%) of our study participants were consuming the recommended levels of fruit and vegetables in a day (≥5 servings) at baseline, and this percentage declined by one-third over 12 months. Of note, there is strong evidence from systematic reviews and meta-analyses that low fruit and vegetable consumption is linked to increased risk for CVDs, cancers, and all-cause mortality [27]. Based on data from epidemiological studies, an increase in BMI of 0.15 kg/m^2^ in this working population over 12 months would translate to a 4% increase in diabetes risk [28] and a 1% increase in CVD risk [29]. A recent systematic review of 50 studies found that overweight and obesity results in high indirect costs to employers due to absenteeism (time away from work), presenteeism (reduced productivity at work), disability, and premature mortality. The excess costs of overweight and obesity due to time away from work alone were estimated to be from US$54 to US$161 and US$89 to US$1586 per annum, respectively [30]. Although the increase in the risk of diabetes and CVDs appears to be small, if the risk factors continue to worsen, then potentially the disease risks would also increase substantially over the longer-term. 

Our study has several strengths. To our knowledge, this is the first study to examine the longitudinal changes in NCD risk factors among employees in workplace settings in Singapore. The study is also unique as it included workers from a variety of occupational settings, including control rooms, workshops, and offices, which are diverse in terms of job types, thereby advancing the generalizability of the study findings. Other strengths include the use of standardized questionnaires and high levels of data completeness (<1% missing for variables). There are certain limitations in our study that need to be acknowledged. Our study sample size was relatively small, which might have reduced the power to detect changes in certain risk factors. There was a 28% loss to follow-up at 12 months. However, this rate is similar to that observed at one year in other workplace cohort studies that have been conducted in Asian settings [31,32,33,34]. This highlights the difficulty in retaining study participants in workplace settings, owing to increased turnover and workers’ busy schedules. There were significant differences in certain baseline characteristics between those followed-up and lost to follow-up, resulting in a possibility of selection bias. The cohort was followed-up only for a short period (i.e., 12 months), and the long-term changes in risk factors are not known. We could not examine the burden and changes in biochemical risk factors as blood tests were not performed at all three time points. However, given the positive correlation (although moderate) between anthropometry and glucose and lipid profiles [35], it could be speculated that some of the biochemical risk factors may also have worsened in this population.

## 5. Conclusions

Rapid urbanization and industrialization have been associated with unhealthy dietary habits and reduced physical activity globally. Consequently, people have become increasingly susceptible to a variety of chronic diseases. A high burden of behavioural and clinical NCD risk factors and a significant worsening of certain key risk factors within a short period of time in this workplace cohort have important implications for the ongoing workplace health and wellness programs that are run by the Health Promotion Board of Singapore [9]. There is a need for innovative and targeted strategies that focus on increasing physical activity and promoting healthy dietary habits of workers with more frequent preventive health checks to raise their awareness and intervene early. Furthermore, our study findings call for continuous surveillance of NCD risk factors in workplace settings and define achievable goals to control the growing NCD epidemic in Singapore.

## Figures and Tables

**Figure 1 ijerph-16-04551-f001:**
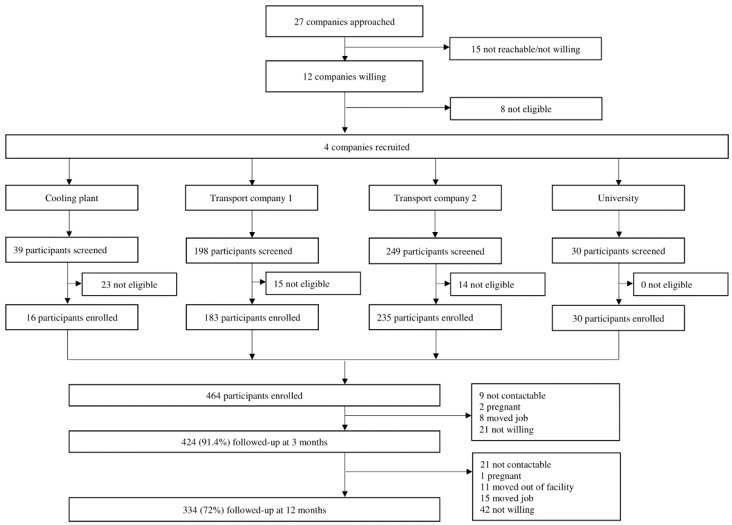
Participant enrolment and follow-up flowchart.

**Table 1 ijerph-16-04551-t001:** Socio-demographic and work-related characteristics of the participants at baseline.

Characteristics	N = 464
Age (years), n (%)	
21–30	153 (33.0)
31–40	121 (26.1)
>40	190 (40.9)
Men, n (%)	369 (79.5)
Ethnicity, n (%)	
Chinese	296 (63.8)
Malays	99 (21.3)
Indians	48 (10.3)
Others ^a^	21 (4.5)
Education, n (%)	
Primary and secondary	49 (10.6)
Pre-college	250 (53.9)
College and above	165 (35.6)
Monthly income, n (%)	
<S$4000	331 (71.3)
≥S$4000	133 (28.7)
Marital status, n (%)	
Single ^b^	184 (39.7)
Married	280 (60.3)
Years employed in the current company, median (IQR)	3 (1–7)
Work location, n (%)	
Underground	142 (30.6)
Aboveground	322 (69.4)
Job type, n (%)	
Control room staff	140 (30.2)
Office staff	225 (48.5)
Workshop staff	99 (21.3)
Job category, n (%)	
Managers and admin personnel	112 (24.1)
Engineers, technicians, and traffic controllers	334 (72.0)
Customer service	18 (3.9)
Work hours/week, mean (SD)	42.8 (6.4)
Shift work (yes), n (%)	166 (35.8)

IQR, inter-quartile range; SD, standard deviation. ^a^ Includes mixed ethnicities, Indonesians, Pakistanis, and Filipinos. ^b^ Includes never married, widowed, divorced, and separated.

**Table 2 ijerph-16-04551-t002:** Behavioural and clinical risk factors for non-communicable diseases at baseline.

Risk factors	N = 464
*Behavioural risk factors*	
Current smoker ^a^, n (%)	113 (24.4)
Alcohol use (in the last 12 months) ^b^, n (%)	248 (53.5)
Total minutes of physical activity/week, median (IQR)	375 (120–900)
Physically inactive ^c^, n (%)	107 (23.1)
Sedentary behaviour (hours/day) ^d^, mean (SD)	6.7 (3.7)
Fruit and vegetable servings/day ^e^, median (IQR)	3.6 (2.2–5.6)
<5 servings of fruit and vegetables/day, n (%)	306 (66.0)
AHEI-2010 score, mean (SD)	22.7 (8.3)
*Clinical risk factors*	
Weight (kg), mean (SD)	72.8 (17.2)
BMI (kg/m^2^), mean (SD)	25.6 (5.2)
Overweight or obesity ^f^, n (%)	311 (67.0)
Central obesity ^g^, n (%)	160 (34.5)

IQR, inter-quartile range; SD, standard deviation; AHEI, Alternate Healthy Eating Index; BMI, body mass index. ^a^ Currently smoking any tobacco products (cigarettes, beedies, cigars, or hookah). ^b^ Alcohol users were those who had at least one standard drink of alcohol (30 mL of spirits, 285 mL of beer, or 120 mL of wine) in the last 12 months. ^c^ Not meeting the World Health Organization’s recommendation of at least 150 min of moderate physical activity per week or at least 75 min of vigorous physical activity per week. ^d^ Time spent (hours) sitting or reclining in a day. ^e^ One serving of fruit and vegetables equals 80 g. ^f^ Body mass index ≥23 kg/m^2^. ^g^ Waist-to-hip ratio ≥0.90 in men and ≥0.85 in women.

**Table 3 ijerph-16-04551-t003:** Age-adjusted changes in behavioural and clinical risk factors for NCDs from baseline to three and 12 months.

**Continuous variables**	**Age-adjusted mean change (95% CI) from baseline to three months**	***p***	**Age-adjusted mean change (95% CI) from baseline to 12 months**	***p***
AHEI-2010 score	0.60 (−0.26 to 1.47)	0.17	0.78 (−0.18 to 1.73)	0.11
Fruit and vegetable servings/day ^a^	−0.12 (−0.37 to 0.12)	0.32	−0.12 (−0.47 to 0.24)	0.52
Total minutes of physical activity/week	−240.7 (−407.4 to −74.1)	0.001	−290.3 (−459.7 to −120.8)	<0.001
Sedentary behaviour (hrs/day) ^b^	−0.20 (−0.51 to 0.10)	0.19	0.10 (−0.21 to 0.42)	0.53
Weight (kg)	0.23 (0.03 to 0.43)	0.024	0.50 (0.25 to 0.75)	<0.001
BMI (kg/m^2^)	0.08 (0.01 to 0.14)	0.031	0.15 (0.06 to 0.23)	0.001
**Categorical variables**	**Odds ratio (95% CI)**	***p***	**Odds ratio (95% CI)**	***p***
Current smoker ^c^	0.92 (0.85 to 1.01)	0.07	1.03 (0.94 to 1.13)	0.55
Alcohol use (in the last 12 months) ^d,^*	--	--	0.86 (0.73 to 1.00)	0.05
<5 servings of fruit and vegetables/day	1.14 (0.91 to 1.44)	0.25	1.33 (1.03 to 1.73)	0.030
Physically inactive ^e^	1.40 (1.10 to 1.79)	0.006	1.64 (1.25 to 2.14)	<0.001
Overweight or obesity ^f^	1.01 (0.94 to 1.09)	0.77	1.15 (1.02 to 1.29)	0.018
Central obesity ^g^	1.05 (0.91 to 1.21)	0.49	1.14 (0.96 to 1.35)	0.13

NCD, non-communicable disease; CI, confidence interval; AHEI, Alternate Healthy Eating Index; BMI, body mass index. ^a^ One serving of fruit or vegetables equals 80 g. ^b^ Time spent (hours) sitting or reclining in a day. ^c^ Currently smoking any tobacco products (cigarettes, beedies, cigars, or hookah). ^d^ Alcohol users were those who had at least one standard drink of alcohol (30 mL of spirits, 285 mL of beer, or 120 mL of wine) in the last 12 months. ^e^ Not meeting the World Health Organization’s recommendation of at least 150 min of moderate physical activity per week or at least 75 min of vigorous physical activity per week. ^f^ Body mass index ≥23 kg/m^2^. ^g^ Waist-to-hip ratio ≥0.90 in men and ≥0.85 in women. * Because of the timeframe for alcohol use (last 12 months), data for the change from baseline to three months could not be calculated.

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
