# Peer review of "Risk Factors for Non-Communicable Diseases at Baseline and Their Short-Term Changes in a Workplace Cohort in Singapore"

_ijerph, 2019, doi:10.3390/ijerph16224551_

Round 1
Reviewer 1 Report
Please check the participants' characteristics such as education level and ethnicity. It is different from the result of your previous work, ref number, 12.
You followed up on your cohort data. Please put a figure about data follow up even you made it in your previous study. You need to put the number of subjects in 3 months and in 12 months.
Please make the same form of variables in table 2 and table 3. You put 'Total minutes of physical activity/week' in table 3, not in table2.
You need to explain the change of 'Total minutes of physical activity/week', 'Waist-to-hip ratio', 'Physically inactive' in the discussion section with reference to previous studies.
Author Response
Reply to reviewers’ comments
Reviewer 1
Comment 1
Please check the participants' characteristics such as education level and ethnicity. It is different from the result of your previous work, ref number, 12.
Reply
Thank you for pointing out the errors. We have now corrected the figures for education and ethnicity in table 1.
Comment 2
You followed up on your cohort data. Please put a figure about data follow up even you made it in your previous study. You need to put the number of subjects in 3 months and in 12 months.
Reply
Thank you for your comment. Figure 1 was already included in the originally submitted version of the manuscript in line 78-79. This figure has details on the number of participants followed-up at three and 12 months. This figure was submitted as a separate file while submitting the manuscript. However, it looks like that this file was not included in the pdf file that was sent for peer review.
Comment 3
Please make the same form of variables in table 2 and table 3. You put 'Total minutes of physical activity/week' in table 3, not in table2.
Reply
Thank you for your comment. We have now included the same form of variables in tables 2 and 3.
Comment 4
You need to explain the change of 'Total minutes of physical activity/week', 'Waist-to-hip ratio', 'Physically inactive' in the discussion section with reference to previous studies.
Reply
Thank you for your comment. In the discussion section, we have now compared the change in physical activity levels observed in our study with findings from previous cohort studies and repeat cross-sectional surveys conducted in the Asian region. This can be found in line 272-283. The magnitude of change in waist-to-hip ratio observed in our study (+0.005 over one year) is clinically not meaningful. Furthermore, change in this measure is not frequently reported in the literature. Therefore, we have now removed this measure from our study.
Reviewer 2 Report
Dear Authors,
Your article on behavioural and clinical risk factors for non-communicable diseases (NCDs) at baseline and their changes over 12 months in a workplace cohort in Singapore has a interesting work and I have some comments that authors should address before publication in the Journal.
Line 81-83: I suggest to explain because the recluted companies not belong to the same product sector
Line 87-88: Why did participants have not travelled overseas across a different time zone at least once a month over the past six months ? I suggest to explain why this information in the criteria is important.
Line 224-226: I suggest to eliminate this phase and to write an phrase of introduction on risk factors
Line 246-248: For my opinion, this phrase is very good. I suggest to expand this topic and explain better the relationship between workers that are overweight or obese and productivity. This is an innovation interpretation of problem.
Author Response
Reply to reviewers’ comments
Reviewer 2
Comment 1
Line 81-83: I suggest to explain because the recluted companies not belong to the same product sector
Reply
Thank you for your comment. We have now amended the text in line 113-117 as below.
A total of 27 companies from a variety of sectors (transport, banks, universities, learning centres, mail service centres, libraries, cooling plants and hospitals) in Singapore with underground workspaces were identified by online searches and collaborator referrals.
Comment 2
Line 87-88: Why did participants have not travelled overseas across a different time zone at least once a month over the past six months ? I suggest to explain why this information in the criteria is important.
Reply
Thank you for pointing this out. The data for this analysis comes from a cohort study that was established primarily to investigate the effects of working in underground spaces on sleep quality and melatonin levels. One of the eligibility criteria for the cohort study was that the participants should not have not travelled overseas across a different time zone at least once a month over the past six months. Frequent travel across different time zones may result in circadian disruption, which could potentially influence sleep and melatonin concentrations. We have now included the below text in line 126-130 to provide greater clarity on this aspect of the eligibility criteria.
The final component of the eligibility criteria is related to the primary objective of the cohort study, which was to study the effects of working in underground spaces on sleep quality and melatonin levels. Frequent travel across different time zones could potentially influence these outcomes due to circadian disruption.
Comment 3
Line 224-226: I suggest to eliminate this phase and to write an phrase of introduction on risk factors
Reply
Thank you for your comment. We have now removed that phrase and replaced it with the below text in line 263-266.
Most NCDs are due to four key behavioural risk factors (tobacco use, harmful use of alcohol, low physical activity and unhealthy diet) and four key clinical risk factors (high blood pressure, high blood glucose, overweight or obesity and high cholesterol.
Comment 4
Line 246-248: For my opinion, this phrase is very good. I suggest to expand this topic and explain better the relationship between workers that are overweight or obese and productivity. This is an innovation interpretation of problem.
Reply
Thank you for your comment. We have now explained the relationship between workers that are overweight or obese and productivity as below in line 302-308.
A recent systematic review of 50 studies found that overweight and obesity results in high indirect costs to employers due to absenteeism (time away from work), presenteeism (reduced productivity at work), disability and premature mortality. The excess costs of overweight and obesity due to time away from work alone were estimated to be from US$54 to US$161 and US$89 to US$1586 per annum, respectively.
Round 2
Reviewer 1 Report
I get the message from the authors that the figure for data collection was included in the paper. However, I failed to find the figure in the main text and supplementary file. Please check the final pub file whether it includes your figure or not.